# Anti-Bacterial and Anti-Inflammatory Properties of Sophoridine and Its Effect on Diarrhea in Mice

**DOI:** 10.3390/ijms26052122

**Published:** 2025-02-27

**Authors:** Jiaxue Wang, Hui Tao, Qiuyu Fan, Zhenlong Wang, Bing Han, Xiumin Wang, Jingquan Wang

**Affiliations:** 1Institute of Feed Research, Chinese Academy of Agricultural Sciences, Beijing 100081, China; 19904762097@163.com (J.W.); taohui@caas.cn (H.T.); fanqiuyu0821@163.com (Q.F.); wangzhenlong02@caas.cn (Z.W.); hanbing02@caas.cn (B.H.); 2Key Laboratory of Feed Biotechnology, Ministry of Agriculture and Rural Affairs, Beijing 100081, China; 3College of Grassland Agriculture, Northwest A&F University, Xianyang 712100, China

**Keywords:** sophoridine, anti-bacterial activity, anti-inflammatory activity, acute intestinal inflammation

## Abstract

To investigate the anti-bacterial and anti-inflammatory properties of sophoridine and elucidate its mechanism of action, we carried out both in vitro and in vivo experiments. Multiple bacterial strains were utilized to determine the effective concentration of sophoridine in antibacterial and bactericidal assays. Subsequently, LPS-stimulated RAW264.7 cells and *E. coli*-challenged BALB/c mice models were employed to evaluate the production of inflammatory cytokines. Our results showed that sophoridine concentrations exceeding 5.12 mg/mL significantly inhibited cell viability, while 0.32 mg/mL of sophoridine demonstrated the optimal anti-inflammatory activity at 12 h. In *E. coli*-induced diarrheal mice, doses of 15, 30, and 60 mg/kg BW of sophoridine alleviated fecal occult blood and exhibited anti-inflammatory effects by reducing the level of serum TNF-α, IL-1β, and IL-6 levels, increasing serum IL-10, and inhibiting leucocyte infiltration in the duodenum. Notably, 15 mg/kg BW of sophoridine effectively decreased the mRNA and protein expression of NF-κB p65. These findings suggest that sophoridine has promising potential for the treatment of diarrhea through its anti-inflammatory effects mediated by the inhibition of NF-κB activation.

## 1. Introduction

In recent years, the hidden dangers of drug residues, increasing resistance, environmental pollution, and potential public health and safety issues caused by antibiotic abuse have promoted the reduction and replacement of antibiotics in China; the exploration of antibiotic alternatives would benefit health and reduce disease with fewer side effects, for instance, plant functional components with anti-bacterial and anti-inflammatory activity, which play an important role in replacing and reducing antibiotics due to their safety, efficacy, stability, and controllability.

Diarrhea is a common global issue, which causes severe harm to human health [1], and the occurrence of diarrhea is highly related to pathogenic microbes [2] and farm animal production [3,4,5]. Pathogenic microbes change the intestinal environment and cause intestinal damage by eliciting inflammation [6]. Instead of using antibiotics, plant active components might be a better choice to improve the health of human and animals [7].

Plant phenolics, terpenoids, alkaloids, lectins, and polypeptides showed antimicrobial and anti-inflammatory properties in vitro and in vivo [8,9,10,11,12,13,14]. Plant phenolic acid, lignan, stilbene, tannin, essential oil, alkaloid, flavonoid, terpene, and plant extract were related to antibacterial activity in the publications collected by Pubmed, and the antibacterial mechanisms of plant ingredients include cell wall and membrane rupture, protein and DNA synthesis disturbance, intracellular ATP depletion, DNA segregation and cell division fimbriae production reduction, metabolic pathways disruption, and so on [7]. Nuclear factor-κB (NF-κB) is a transcription factor that plays in a crucial role in regulating the inflammatory response; activated NF-κB p65 has been implicated in various inflammatory diseases, such as arthritis [15,16], colitis [17], neuroinflammation [18], periodontal inflammation [19], renal epithelial inflammation [20], or skin inflammation [21]. Diarrhea is usually accompanied by inflammation; inhibition of inflammation can effectively attenuate intestinal damage.

Sophoridine is one of the quinolizidine-based alkaloids identified as active components in *Sophora flavescens* [22], *Sophora alopecuroides* L. [23], the flower of *Sophora japonica* Linn [24], *Euchresta japonica* Benth, and *Sophora moocrorftinan* [25], and it was reported that the sophoridine has anticancer [26,27,28,29], anti-inflammatory [30], antiviral activity [31], and so on. However, to date, no studies have investigated the effects of sophoridine on bacterial diarrhea and its underlying mechanisms.

In the present study, we investigated the potential of sophoridine to inhibit bacterial growth and reduce inflammatory damage in mice. This may provide promising insights into the application of plant active ingredients as alternatives to antibiotics in future therapeutic strategies.

## 2. Results

### 2.1. Antibacterial Activity of Sophoridine

The minimum inhibitory concentration (MIC) and minimum bactericidal concentration (MBC) of sophoridine against *Escherichia coli* CACC1515, *E. coli* CVCC195, and *Salmonella typhimurium* ATCC14028 were 5.12 mg/mL and 10.24 mg/mL, respectively. The MIC of sophoridine against *S. enteritidis* CVCC3377 and *Staphylococcus aureus* ATCC43300 was 10.24 mg/mL. Sophoridine had better antimicrobial activity against *E. coli* CACC1515, *E. coli* CVCC195, and *S. typhimurium* ATCC14028 than that against *S. enteritidis* CVCC3377 and *S. aureus* (Table 1).

### 2.2. Cytotoxicity of Sophoridine

Sophoridine at a concentration higher than 5.12 mg/mL had a significantly inhibitory effect on cell viability compared to the control group (*p* < 0.05). Specifically, 5.12 mg/mL and 10.24 mg/mL of sophoridine significantly inhibited cell viability (*p* < 0.05), while 0.32 mg/mL of sophoridine significantly promoted cell proliferation (*p* < 0.05). Moreover, sophoridine at a concentration lower than 2.56 mg/mL had no toxic effect on cells (Figure 1).

### 2.3. Effect of Sophoridine on NO Release

Compared with the blank control group, the LPS-challenged groups showed significantly higher relative nitric oxide (NO), except for those in the H0.32 group, H0.16 group, and H0.08 group at 3 h, 12 h, and 24 h (*p* < 0.05). Compared with the positive control group, the sophoridine-treated groups showed significantly lower relative NO in the S0.08 group, S0.16 group, and S0.32 group at 12 h and 24 h (*p* < 0.05). The S0.32 group had no significant difference from the BC group at 12 h, which means 0.32 mg/mL of sophoridine had the best anti-inflammatory activity at 12 h but did not have significant anti-inflammatory activity in the cell culture at 24 h (Figure 2).

### 2.4. Fecal Occult Blood Score

The mice were examined, and the possible bleeding caused by intestinal inflammation was confirmed through fecal occult blood testing. The fecal occult blood was the most severe on d3, for which the score reached 3+ (Table 2). When the mice were treated with sophoridine, the fecal occult blood alleviated and returned to normal at d6 (Table 2), similar to the control group, and high-dose sophoridine was as effective as chloramphenicol hydrochloride in reducing the symptom of fecal occult blood.

### 2.5. Serum Inflammatory Cytokines of Mice

Compared with the M group, the sophoridine at dose of 15, 30, or 60 mg/kg BW significantly decreased the levels of serum tumor necrosis factor-α (TNF-α), interleukin-1β (IL-1β), and interleukin-6 (IL-6) and increased the level of interleukin-10 (IL-10) significantly (*p* < 0.05). Specifically, sophoridine at 15 mg/kg BW decreased TNF-α, and sophoridine at 30 mg/kg BW decreased IL-1β to the level of the C group. Although sophoridine inhibited serum IL-6, its effect was not as potent as that of the P group. However, it increased serum IL-10 to the level of the P group (Figure 3).

### 2.6. Pathological Changes in Mouse Duodenum

As shown in Figure 4, the duodenum in the C group and HS group exhibited similar pathological changes, characterized by distinct and intact villi and glands, accompanied by minimal inflammatory cell infiltration (indicated by the black arrow). In contrast, although the P group also presented clear and intact villi, occasional inflammatory cell infiltration was observed. Notably, the M group showed shortened villi, prominent inflammatory cell infiltration, as well as swollen and thickened glands (marked by blue arrow). The histological score is shown in Table 3, and the total score of the M group reached 5, whereas the score of the HS group was considerably lower than that of the M group.

### 2.7. Effect of Sophoridine on mRNA Expression of NF-κB p65

As shown in Figure 5, the mRNA expression of NF-κB p65 was significantly higher in the M group than that in the C, P, LS, MS, and HS groups (*p* < 0.05). Additionally, the mRNA expression of NF-κB p65 was significantly lower in the P, LS, MS, and HS groups than in the C group (*p* < 0.05). However, there was no significant difference in the mRNA expression of NF-κB p65 among the P, LS, MS, and HS groups.

### 2.8. Effect of Sophoridine on Protein Expression of NF-κB p65 and Phosphorylated NF-κB p65

The protein expression levels of NF-κB p65 and phosphorylated NF-κB p65A were determined by Western blot. As shown in Figure 6, a significant difference was observed in the protein expression level of NF-κB p65 but not in that of phosphorylated NF-κB p65. Mice treated with a low concentration (15 mg/kg BW) of sophoridine had lower NF-κB p65 protein expression, reaching the same level as in the C group and the P group. However, sophoridine at 30 or 60 mg/kg BW did not decrease the NF-κB p65 protein expression in the duodenum, and the protein expression level was similar to that in the M group.

## 3. Discussion

It was reported that the MIC of sophoridine against *E. coli* was 2 × 10^−2^ mol/L [32], equivalent to 20 mmol/L. This value is similar to that obtained in the present study, where the MIC of sophoridine against *E. coli.* was 5.12 mg/mL (20.6 mmol/L). Sophoridine also demonstrated antibacterial activity against *S. enteritidis* (MIC = 5.24 mg/mL), *S. typhimurium* (MIC = 10.24 mg/mL), and *S. aureus* (MIC = 10.24 mg/mL). Moreover, some quinolizidine-type alkaloids exhibit better antibacterial activity (with MIC values ranging from 8 to 208.3 μg/mL) in comparison [33]. Sophoridine is structurally similar to matrine, of which the antibacterial activity involved the introduction of hydrophilic groups disrupting the bacterial cell membrane [34] or inhibiting *E. coli* biofilm formation [35]. In our previous study, we found that sophoridine could inhibit the GTPase activity and FtsZ assembly of *E. coli*, which further inhibits the amplification of *E. coli.* and kills *E. coli* [36]. It could potentially be one of the antibacterial mechanisms of sophoridine; however, more research is necessary to reveal its exact antibacterial mechanism.

In our study, sophoridine significantly inhibited cell viability at concentrations above 5.12 mg/mL (or 20.62 mM). However, it exhibited no cytotoxicity towards cells at concentration below 2.56 mg/mL (or 10.31 mM) and significantly promoted cell proliferation at a concentration of 0.32 mg/mL (or 1.29 mM) (Figure 1). It was reported that sophoridine showed no cytotoxicity against HKC and LX-2 cell lines at a high concentration of 160 μM (or 0.040 mg/mL) [27], but it inhibited the activity of rat liver cells at the IC_50_ value of 1.9 mM (or 0.47 mg/mL) at 24 h [37]. Additionally, it significantly induced cytotoxicity of D283-Med cells at concentration of 1 or 2 mg/mL [38] and mild cytotoxicity of Vero cells at concentrations less than 1 mg/mL [31]. Moreover, it significantly and dose-dependently suppressed the growth of HepG2-LR and Huh7-LR cells at 20, 40, and 80 μmol/L [39]. Notably, the concentrations of sophoridine in these previous studies were much lower than that in the present study. It was reported that sophoridine inhibited the growth of human pancreatic, gastric, liver, colon, gallbladder, and prostate carcinoma cells, with an IC_50_ value in the range of 20–200 μmol/L. In contrast, normal human pancreatic ductal epithelial cells and bronchial epithelial cells were less sensitive to sophoridine concentrations in the range of 0–500 μmol/L [40]. The result of our cell viability assays did not corroborate those of previous studies, which might be attributed to the use of different cell lines in the respective studies.

Nitric oxide (NO), a cellular signaling molecule produced by NO synthases (NOS) in cells or tissues, plays a crucial role in the pathogenesis of inflammation. It acts as a pro-inflammatory mediator when excessive amounts are produced and a toxic defense molecule against infectious organisms [41]. Previous reports indicated that LPS stimulated RAW 264.7 and THP-1 cells to produce more NO. Moreover, 20 μg/mL and 40 μg/mL of sophoridine also led the cells to release more NO compared to the control group [29], which is inconsistent with our findings. We found that sophoridine-treated groups, specifically the S0.08, S0.16, and S0.32 groups, showed significantly lower relative NO production at 12 h and 24 h (*p* < 0.05). The S0.32 group showed no significant difference from the blank control (BC) group at 12 h, suggesting that it might represent the optimal dosage and time point for inhibiting inflammation in vitro. It has been reported that NO production is related to the expression of inducible NOS, and the inhibitors of NO production or inducible nitric oxide synthase (iNOS) have therapeutic effects on inflammation [42,43]. Given that sophoridine is an inhibitor of both NO and iNOS, it could potentially be a therapeutic agent for managing inflammation.

Fecal occult blood is one of the stool markers used to detect and monitor intestinal inflammation and damage [44]. Plant-derived active ingredients have demonstrated positive effects in alleviating and treating diarrhea and intestinal damage [45,46,47,48]. Sophoridine has a structure similar to that of matrine. It has been found that matrine alleviated intestinal barrier damage in diarrhea-induced mice infected by parasites, including reducing intestinal bleeding [49]. This is consistent with our results, which show that low, middle, or high doses of sophoridine reduced fecal occult blood in mice after 6 days of treatment compared to the model group. Moreover, a high dose of sophoridine (60 mg/kg BW) had a therapeutic effect similar to chloramphenicol hydrochloride.

It has been reported that ibuprofen reduces TNF-α but does not affect IL-6, while 48.6 mg/kg BW or 12.15 mg/kg BW of sophoridine has no influence on TNF-α but increases IL-6 in the local inflammatory exudates of a carrageenin-induced mouse paw edema model [30], which is consistent with our findings that chloramphenicol hydrochloride decreased serum TNF-α and did not affect serum IL-6, and sophoridine increased serum IL-6, but 30 mg/kg BW or 60 mg/kg BW of sophoridine decreased serum TNF-α in diarrhea mice compared to the control group. Our result also aligned with the observation that sophoridine decreased serum TNF-α and IL-6 in cachectic mice compared to the LPS-challenged mice [50]. IL-1β is one of the pro-inflammatory cytokines involved in the host’s inflammatory response to pathogens, and it could exacerbate tissue damage and lead to diseases if unrestrained [51,52]. Sophoridine significantly reduced serum IL-1β; specifically, 30 mg/kg BW of sophoridine decreased the serum IL-1β to the level of the C group, while 60 mg/kg BW of sophoridine reduced it to the level of the P group. This indicates that high concentrations of sophoridine can inhibit IL-1β, similar to some antibiotics. IL-10, an important anti-inflammatory cytokine produced by macrophages, plays a crucial role in maintaining gastrointestinal homeostasis [53]. It has been reported that a lack of IL-10 can exacerbate intestinal epithelial injury induced by non-steroidal anti-inflammatory drugs (NSAIDs) [54]. We found that serum IL-10 significantly increased in the P, LS, MS, and HS groups, which suggests that antibiotic (P group) or sophoridine (LS, MS, HS groups) can repair intestinal injury to a certain extent by increasing IL-10. IL-10 production is regulated by dietary factors rather than gut microbiota in the small intestine compared to the colon [55]. In our study, mice challenged with *E. coli.* did not show an increase in serum IL-10 in the M group compared to the C group, but sophoridine could increase it. It was reported that sophoridine could enhance autophagy of macrophages and reduce secretion of inflammation factors in acute lung injury induced by LPS [56]. Overall, sophoridine exhibited anti-inflammatory effects by decreasing serum TNF-α, IL-1β, and IL-6 and increasing serum IL-10. Thus, sophoridine might serve as a functional component in the recovery from intestinal inflammation.

Inflammation response is related to the migration of immune cells [57,58], and we found severe infiltration of inflammatory cells in the M group but little infiltration in the C and HS groups (Figure 4), which suggested that sophoridine inhibited leucocytes’ infiltration into the inflammatory site, which might be one of the mechanisms of its anti-inflammatory effect.

Sophoridine has been shown to inhibit the expression of NF-κB p65 in renal tissue in the level of mRNA and protein [59]. It also blocks the NF-κB pathway by reducing the expression of NF-κB, thereby alleviating lung injury in mice [60]. Additionally, it down-regulates NF-κB protein expression in pancreatic cancer cells [61] and the expression of NF-κB p65 in liver cancer cells [62]. In the present study, we observed that sophoridine decreased the expression of NF-κB at both in the mRNA level (with 15, 30, 60 mg/kg BW of sophoridine) and the protein level (with 15 mg/kg BW of sophoridine), which indicates that 15 mg/kg BW of sophoridine can inhibit intestinal inflammation by reducing NF-κB. The phosphorylation of NF-κB p65 increases its intramolecular flexibility, expanding and specifying the repertoire of possible protein–protein interactions [63]. It has been reported that sophorolipid significantly prevents NF-κB p65 from translocating from the cytoplasm to the nucleus and inhibits the phosphorylation of NF-κB p65, which in turn further reduces the inflammatory response [64]. However, in the present study, no significant difference in phosphorylated NF-κB p65 was found between the groups, which suggests that the inhibition of intestinal inflammation induced by *E. coli* may be less related to the phosphorylation of NF-κB p65.

## 4. Materials and Methods

### 4.1. Chemicals, Drugs, Cells, and Animals

The following were procured: sophoridine (C15H24N2O, 98.0% pure), a monomeric alkaloid extracted from the Sophora plant in the *Fabaceae* family (Lemeitian Med. Inc., Chengdu, China); lipopolysaccharide (LPS), LPS L8880 from *E. coli* (O55:B5) (Solarbio Co., Ltd., Beiijng, China); RAW264.7 cells (American Type Culture Collection, Manassas, VA, USA); *S. typhimurium* ATCC14028, *S. enteritidis* CVCC3377, *S. aureus* ATCC43300, *E. coli* CVCC1515, and *E. coli* CVCC195 (Solarbio Co., Ltd., Beijing, China); BALB/c male mice, six weeks of age, 20 ± 2 g BW (Vital River Co., Ltd., Beijing, China).

### 4.2. Anti-Microbial Activity of Sophoridine

The anti-microbial activity of sophoridine was determined against different bacteria strains, including *S. typhimurium* ATCC14028, *S. enteritidis* CVCC3377, *S. aureus* ATCC43300, *E. coli* CVCC1515, and *E. coli* CVCC195. The MIC value of sophoridine was measure according to the method reported by Wiegand et al. [65] with some modifications. The bacterial strains were inoculated and grown to mid-log phase in fresh Mueller–Hinton broth (MHB) at 37 °C. Bacterial inoculum suspensions were prepared at a final concentration of approximately 1 × 10^5^ CFU/mL. Then, 90 μL of inoculum suspensions was added to each well of 96-well plates. After the sophoridine was two-fold diluted, from 10.24 to 0.005 mg/mL, 10 μL of each concentration of sophoridine solution was added. A positive control without sophoridine was also included. Broth without bacteria was used as the negative control. The 96-well plates were incubated at 37 °C for 18–24 h until visible turbidity was observed in the positive control wells by visual inspection. At this point, the MIC was determined. The bacterial solution from the clarified wells (show no visible bacterial growth) were taken and plated on Mueller–Hinton agar (MHA) plates. After overnight incubation at 37 °C, the MBC value was determined. Each concentration of sophoridine was tested in 3 biological replicates, and the experiments were conducted at least three times.

### 4.3. Cytotoxicity Evaluation of Sophoridine

The cytotoxicity of sophoridine was evaluated against daughter clones of mouse macrophage cell line, RAW264.7, following a modified method reported by Riss et al. [66]. A cell count of 5 × 10^4^ RAW264.7 cells was seeded in 96-well microplates using DMEM medium supplemented with 10% fetal calf serum (FCS) and 0.1% penicillin/streptomycin for cell maintenance and growth. When the cells approached the end of the exponential growth phase, reaching approximately 70–80% confluent, sophoridine at different concentrations (0, 0.02, 0.04, 0.08, 0.16, 0.32, 0.64, 1.28, 2.56, 5.12, and 10.24 mg/mL) was added to the cells. The cells were then cultured in regular medium without FCS. Each concentration of sophoridine was added to the plate, performed in triplicate. After 24 h, 10% CCK-8 reagent was added, and the plates were incubated for additional 1 h. Subsequently, the optical density (OD) at 450 nm was measured for each well to determine the absorbance. The cell viability was calculated according to the following equation: cell viability (%) = (OD_treatment_ − OD_background control_)/(OD_negative control_ − OD_blank control_) × 100. The protocol was conducted at least three times.

### 4.4. Measurement of Extracellular NO in RAW264.7 Cells Challenged by LPS

RAW264.7 cells were cultured in a 12-well plate with a culture medium supplemented with 15% fetal calf serum (FCS). Once the cells reached confluence, they were cultured in FCS-free regular medium containing 1 μg/mL of LPS except for the blank control group (BC group). Then, 0 (positive control group, PC group), 0.64 (H0.64 group), 0.32 (H0.32 group), 0.16 (H0.16 group), or 0.08 (H0.08 group) mg/mL of sophoridine was added after 12 h. The supernatants of the cell cultures were harvested at 3, 6, 12, and 24 h after being centrifuged at 500× *g* for 5 min, and then the NO concentrations were measured as the absorbance at 570 nm using a NO detection kit (Beyotime Biotechnology, Beijing, China). After lysing and centrifuging the cells, the protein concentration of cells was detected using a BCA protein test kit. The relative yield of NO was calculated according to the following equation: NO relative concentration (μmol/L) = NO concentration/protein concentration.

### 4.5. Trial in Mice

Male BALB/c mice (six weeks of age, 20 ± 2 g BW) were obtained from Beijing Vital River Laboratories Biotechnology Co, Ltd. (Certificate NoSCXK2022-0007, Beijing, China). All experiments were approved by the Laboratory Animal Ethics Committee Feed Research Institute, Chinese Academy of Agricultural Sciences, which were performed in accordance with animal welfare practices and procedures following the Guidelines for Animal Experiments by the Ministry of Science and Technology (2006, Beijing, China).

#### 4.5.1. Administration

Thirty mice were randomly divided into six groups, treated with saline (control group, C group), *E. coli* CVCC1515 (model group, M group), *E. coli* CVCC1515 and 60 mg/kg BW of chloramphenicol hydrochloride (positive control group, P group), *E. coli* CVCC1515 and 15 mg/kg BW of sophoridine (low dose of sophoridine group, LS group), *E. coli* CVCC1515 and 30 mg/kg BW of sophoridine (middle dose of sophoridine group, MS group), and *E. coli* CVCC1515 and 60 mg/kg BW of sophoridine (high dose of sophoridine group, HS group). Except for the mice in the control group, all other mice were challenged by gavage with 200 μL of *E. coli* CVCC1515 (2.5 × 10^9^ CFU/mL), every 24 h for three times total (d1 to d3). Subsequently, chloramphenicol hydrochloride or different concentrations of sophoridine were intragastrically administrated to the mice in the P group, LS group, MS group, or HS group, respectively, every 24 h for three times total (d4 to d6). The mice had free access to food and water and were maintained under standard housing conditions (24–27 °C, RH 60–65%) on a 12 h light/dark cycle.

#### 4.5.2. Fecal Occult Blood Detection

Feces were collected daily for fecal occult blood detection by using a test kit (60403ES60, Yeasen Tec. Inc., Shanghai, China). Here, 0.5 mg of feces was smeared in the grid of the test card, and then the solutions were added according to the manufacturer’s protocol. The minimum reaction time and the color were recorded. A purple–blue or purple–red color indicated the strongest positive reaction, and the severity of occult blood was quantitated in a range of -, +, ++, and +++.

#### 4.5.3. Serum Inflammatory Cytokines Analysis

At the end of the trial, the mice were fasted overnight. Subsequently, they were anesthetized, and blood was collected from the orbital sinus. Blood samples were centrifuged at 2000× *g* and 4 °C for 10 min to obtain the serum. The levels of serum inflammatory cytokines, namely TNF-α, IL-1β, IL-6, and IL-10, were then measured using commercial assay kits (Sinoukbio Ins., Beijing, China) in accordance with the manufacturer’s instructions.

#### 4.5.4. Histological Change in the Duodenum

Immediately after blood sampling, the mice were sacrificed by cervical dislocation. Subsequently, the duodenums were excised. In the C group, M group, and HS group, sections of the duodenum were fixed in 10% phosphate-buffered formalin for subsequent hematoxylin–eosin staining. The slices were visualized under an optical microscope (CX31, Olympus, Tokyo, Japan) at magnifications of 100× and 200×. The histological score of the inflammatory intestine was evaluated according to the method reported by Neri et al. [67].

#### 4.5.5. Expression of NF-κB p65 Assayed by RT-PCR and Western Blot

The duodenum tissue was divided into two parts: one for mRNA expression analysis and the other one for protein expression analysis.

The total RNA of duodenum was extracted using Trizol reagent (Tiangen, Shanghai, China) according to the manufacturer’s protocol. The concentration of RNA was measured by using a UV–VIS spectrophotometer (NanoDrop 2000, Thermo, Waltham, MA, USA), and the quality of RNA was accepted when the value of A 260/A280 ranged from 1.8 to 2.0. The total RNA was reverse transcribed, and real-time quantitative PCR was carried out on the resulting cDNA using SYBR Green I (iCycler iQ Real-Time PCR Detection System; Bio-Rad, Hercules, CA, USA). The primer synthesis of NF-κB p65 was performed by Sangon Biotech Co. Ltd. (Shanghai, China), and the oligonucleotide sequences of the primers were 5′-GATGTTCACGGTGTGACCCTA-3′ and 5′-CATGAGTGCTGGAAGAGCCC-3′ for NF-κB p65 and 5′-CACTGTCGAGTCGCGTCC-3′ and 5′-CGCAGCGATATCGTCATCCA-3′ for β-actin. The mRNA levels of NF-κB p65 were quantified and expressed normalized to β-actin, which served as an internal reference. All experiments were conducted in triplicate, and the gene expression levels were analyzed using the 2^−ΔΔCt^ method [68].

The protein expression of NF-κBp 65 was analyzed according to the method reported by Kurien and Scofield [69]. The duodenum tissue was placed in RIPA buffer and lysed using a handheld homogenizer. The protein concentration of lysates was detected using NanoDrop 2000 (Thermo, Waltham, MA, USA). Equivalent amounts of protein for each sample were loaded onto a 15% sodium dodecyl sulfate polyacrylamide (SDS-PAGE) gel. After the proteins were transferred to a nitrocellulose membrane, the membranes were washed with TBST (ST673, Beyotime Biote. Inc., Shanghai, China) and then blocked using 5% non-fat dried milk diluted in TBST for 2 h. The membranes were incubated overnight with a rabbit monoclonal antibody against NF-κB p65 (ab32523, Bioss, Beijing, China), phosphorylated NF-κB p65 (3033 T, Cell Signaling, Danvers, MA, USA) and β-Actin 13E5 (4970 T, Cell Signaling Technology Inc., Danvers, MA, USA). Afterward, the membranes were washed three times with TBST and incubated with horseradish peroxidase-conjugated secondary antibody for 45 min. The protein bands on the membranes were visualized by adding enhanced chemiluminescence reagents. Image acquisition and densitometric analysis of the selected blots in the membranes were performed using ImageJ software (version 1.8.0).

### 4.6. Statistical Analysis

Data are expressed as means ± standard deviation, and statistical analyses were performed using one-way ANOVA followed by Tukey’s multiple comparisons test on SPSS22.0 (Armonk, NY, USA). The threshold for defining statistical significance is *p* < 0.05.

## 5. Conclusions

Sophoridine at concentrations above 5.12 mg/mL demonstrated potent anti-bacterial activity against several Gram-negative and Gram-positive bacteria in vitro, but sophoridine at concentrations below 2.56 mg/mL was non-cytotoxic. This indicates that sophoridine does not have antidiarrheal effects through anti-bacterial activity in the present study. However, sophoridine at a concentration of 0.32 mg/mL could effectively alleviate the inflammatory reaction by reducing the production of NO in LPS-stimulated RAW264.7 cells. Sophoridine exerted a protective effect on intestinal inflammation by reducing the fecal occult blood, modulating the secretion of inflammatory factors through decreasing pro-inflammatory factors and potentially increasing anti-inflammatory ones. Sophoridine at a concentration of 15 mg/kg BW exhibited promising anti-diarrheal effects in mice challenged with *E. coli* achieved by inhibiting the inflammation response mediated by NF-κB p65. Overall, the results of the study suggest that sophoridine could potentially serve as an antidiarrheal agent in the treatment of intestinal inflammation.

## Figures and Tables

**Figure 1 ijms-26-02122-f001:**
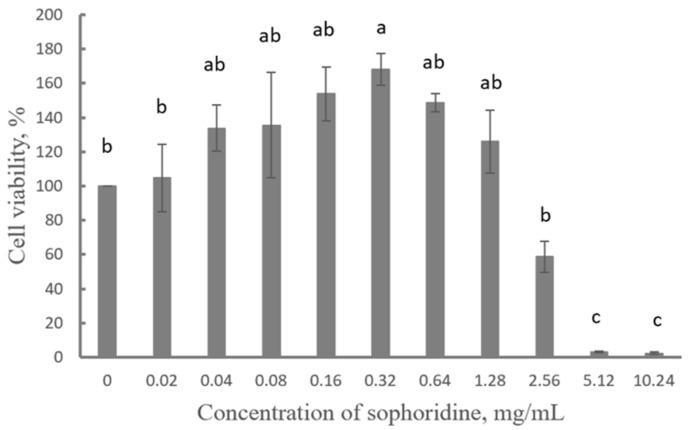
Viability of RAW264.7 cells upon exposure to different concentrations of sophoridine. Different letters stand for significance between groups.

**Figure 2 ijms-26-02122-f002:**
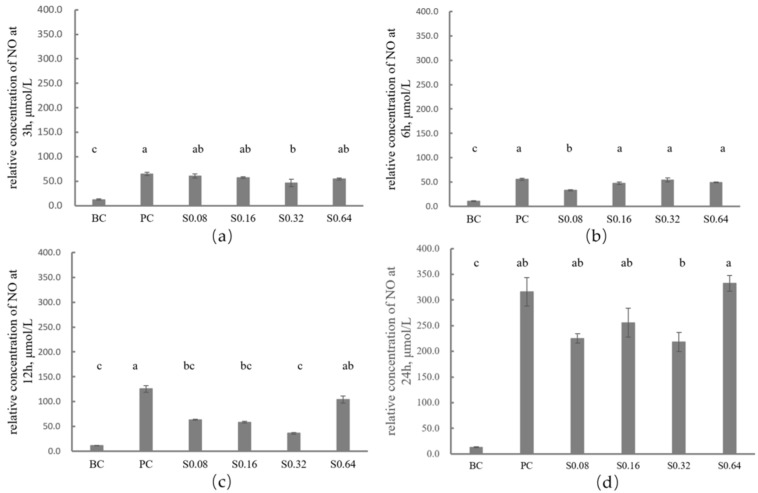
Effects of sophoridine on NO relative concentration at different times in LPS-stimulated RAW264.7 cells. (**a**) Effects of sophoridine on NO relative concentration at 3 h. (**b**) Effects of sophoridine on NO relative concentration at 6 h. (**c**) Effects of sophoridine on NO relative concentration at 12 h. (**d**) Effects of sophoridine on NO relative concentration at 24 h. BC, black control group; PC, positive control group; S0.64, 0.64 mg/mL of sophoridine; S0.32, 0.32 mg/mL of sophoridine; S0.16, 0.16 mg/mL of sophoridine; S0.08, 0.08 mg/mL of sophoridine. Different letters of a, b, and c indicate significant differences between treatments.

**Figure 3 ijms-26-02122-f003:**
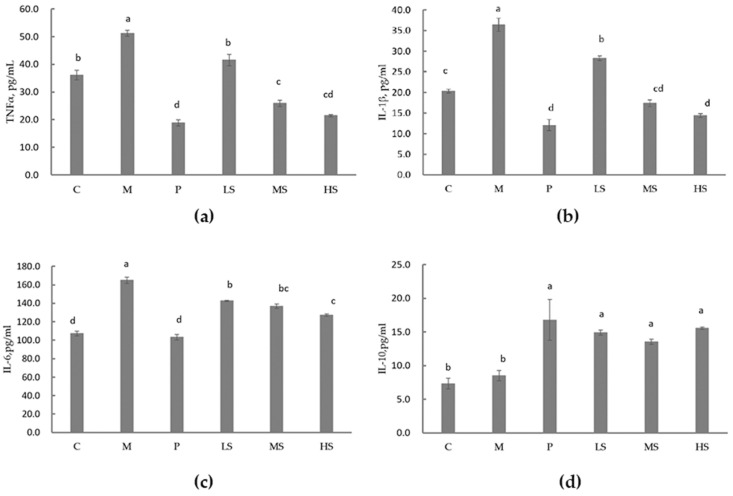
Effect of sophoridine on serum inflammatory cytokines in mice challenged by *E. coli*. (**a**) Effect of sophoridine on TNF-α. (**b**) Effect of sophoridine on IL-1β. (**c**) Effect of sophoridine on IL-6. (**d**) Effect of sophoridine on IL-10. C, control group; M, model group; P, positive group (treated with chloramphenicol hydrochloride); LS, low dose (15 mg/kg BW) of sophoridine group; MS, middle dose (30 mg/kg BW) of sophoridine group; HS, high dose (60 mg/kg BW) of sophoridine group. Different letters stand for significance between groups, the same letters indicate that the results are not significant, and different letters indicate that there are significant differences in the results.

**Figure 4 ijms-26-02122-f004:**
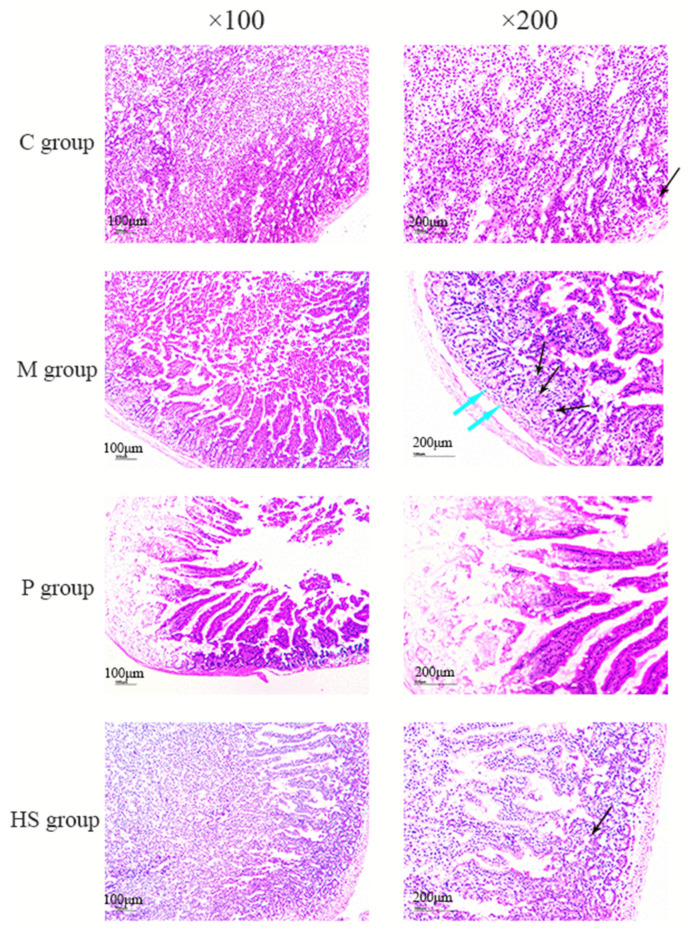
Effect of sophoridine on mouse duodenum challenged by *E. coli*. C group, control group; M group, model group; P group, positive group (treated with chloramphenicol hydrochloride); HS group, high dose (60 mg/kg BW) of sophoridine group. The blue arrows points swollen and thickened glands; the black arrows points inflammatory cell infiltration.

**Figure 5 ijms-26-02122-f005:**
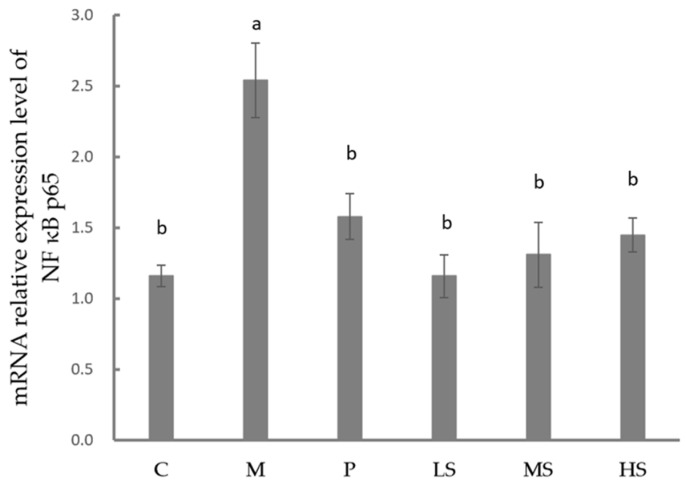
Relative mRNA expression of NF-κB p65 in intestine of mice treated by sophoridine. C, control group; M, model; P, positive group (treated with chloramphenicol hydrochloride); LS, low concentration of sophoridine (15 mg/kg BW) group; MS, middle concentration of sophoridine (30 mg/kg BW) group; HS, high concentration of sophoridine (60 mg/kg BW) group. Different letters stand for significance between groups.

**Figure 6 ijms-26-02122-f006:**
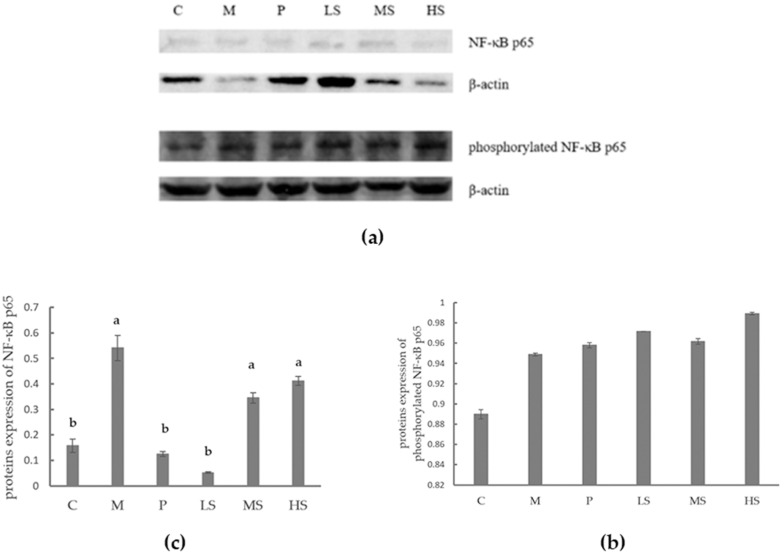
Effect of sophoridine on proteins’ expression of NF-κB p65 and phosphorylated NF-κB p65. (**a**) Western blot analysis of NF-κB p65, phosphorylated NF-κB p65, and β-actin. The quantitative analysis results for the relative signal densities of NF-κB p65 and phosphorylated NF-κB p65 after normalization to the signal density of β-actin are shown in (**b**,**c**). Different letters stand for significance between groups. C, control group; M, model; P, positive group (treated with chloramphenicol hydrochloride); LS, low concentration of sophoridine (15 mg/kg BW) group; MS, middle concentration of sophoridine (30 mg/kg BW) group; HS, high concentration of sophoridine (60 mg/kg BW) group. Different letters stand for significance between groups; the same letters indicate that the results are not significant, and different letters indicate that there are significant differences in the results.

**Table 1 ijms-26-02122-t001:** MIC and MBC of sophoridine against *E. coli*, *S. typhimurium*, *S. enteritidis*, and *S. aureus*.

Item	*E. coli* CACC1515	*E. coli* CVCC195	*S. typhimurium* ATCC14028	*S. enteritidis* CVCC3377	*S. aureus* ATCC43300
MIC, mg/mL	5.12	5.12	10.24	5.12	10.24
MBC, mg/mL	10.24	10.24	>10.24	10.24	>10.24

MIC, minimal inhibit concentration; MBC, minimum bactericidal concentration.

**Table 2 ijms-26-02122-t002:** Fecal occult blood score in mice.

	C	M	P	LS	MS	HS
d 1	-	++	+	+	++	++
d 2	-	++	++	++	++	++
d 3	-	+++	+++	+++	+++	+++
d 4	-	+++	+	++	++	++
d 5	-	++	-	++	+	-
d 6	-	++	-	-	-	-

C, control group; M, model group; P, positive group (treated with chloramphenicol hydrochloride); LS, low dose (15 mg/kg BW) of sophoridine group; MS, middle dose (30 mg/kg BW) of sophoridine group; HS, high dose (60 mg/kg BW) of sophoridine group. “-” means that the fecal was no purple blue or purple red color reaction during the interpretation time; “+” means that purple color is gradually produced within 1–2 min; “++” means that purple-red color is produced within 1 min; “+++” means that purple-blue color was produced within 10 s. The depth of color indicates the severity of occult blood in stool.

**Table 3 ijms-26-02122-t003:** Histological score of the intestine.

Group	Inflammatory Cell Infiltration	Epithelial Change	Villous Integration	Total Score
C	0	0	0	0
M	3	1	1	5
P	0	0	0	0
HS	1	0	0	1

C group, control group; M group, model group; P group, positive group (treated with chloramphenicol hydrochloride); HS group, high dose (60 mg/kg BW) of sophoridine group. The inflammatory cell infiltrate: none-0, mild-1, moderate-2, severe-3; surface epithelial integrity: none-0, mild-1, moderate-2, severe-3; villous integration: none-0, mild-1, moderate-2, severe-3.

## Data Availability

The raw data supporting the conclusions of this article will be made available by the authors on request.

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
