# Peer review of "Anti-Bacterial and Anti-Inflammatory Properties of Sophoridine and Its Effect on Diarrhea in Mice"

_ijms, 2025, doi:10.3390/ijms26052122_

Round 1
Reviewer 1 Report
Comments and Suggestions for Authors
In this manuscript, authors Wang et al evaluated the anti-bacterial and anti-inflammatory Properties of Sophoridine and its effect on diarrhea in mice. Considering the effects of diarrhea on human health and increasing resistance to available antibiotics in Asian countries, this is an important and interesting study that aims to investigate new agents to alleviate bacterial diarrhea using a mouse model system. Experiments are conducted well and results are clearly presented. The study results showed that sophoridine significantly inhibited cell viability, exhibited the ant-inflammatory activity in vitro and in E. coli-induced diarrheal mice, alleviated fecal occult blood and exhibited anti-inflammatory effects. Overall, this is a nice study demonstrating that sophorodine has promising potential for the treatment of diarrhea through its anti-inflammatory effects. However, there are several issues that need to addressed.
1. Why was stool marker (fecal occult) only evaluated in the in vivo study.
2. Why were no comparator controls (e.g. antibiotics) used in the in vitro or in vivo studies?
3. Please provide the inflammatory score in the intestines (in vivo studies).
4. Western blot results are not clear. Please provide a good image with multiple samples for each group.
5. The quantitative images representing western blot doesnt seem to match up. Please check.
Author Response
Reviewer 1
Comments 1: Why was stool marker (fecal occult) only evaluated in the in vivo study.
Response1: Thank you for pointing this out. The fecal occult, an indicator of intestinal inflammation status, is invisible to the naked eye and difficult to observe under a microscope, Moreover, it has to be related to experimental animal, the cells we used in the present study can’t produce occult, and antibacterial test and cell experiment can’t be possible to involve bleeding indicator.
Comments 2: Why were no comparator controls (e.g. antibiotics) used in the in vitro or in vivo studies?
Response 2: Thank you for pointing this out. We did include a comparator control in vivo study, and the mice in P group (positive group) were treated with chloramphenicol hydrochloride, which is an amide alcohol antibiotic. Regarding the absence of antibiotics in the in-vitro study. Let me clarify, firstly, we need to find out the inhibit and bactericidal concentration of sophoridine; subsequently, we had to find the safe concentration of sophoridine for cells, which were then used to determine the appropriate sophoridine concentration for later animals’ trials. In the study, since we investigated the potential of sophoridine as an antidiarrheal agent, we compared the effects of sophoridine with those of antibiotics in the in-vivo study.
Comments 3: Please provide the inflammatory score in the intestines (in vivo studies).
Response 3: Thank you for pointing it out. We had the histological score of the intestines supplied in Line 148-154 and Line 389-390 of the revised manuscript.
Comments 4: Western blot results are not clear. Please provide a good image with multiple samples for each group.
Response 4: Thank you for your advance. We detected the protein expression of NFκB-p65 and phosphorylated NFκB-p65 several times. However, it seemed that the expression of NFκB-p65 was not that particularly high. The images of samples presented in the manuscript were from the first run of the samples, and we still decided to use the image in the manuscript. The replicates of the samples were provided in the ‘Figures, Graphics, Images’ along with the revised manuscript in the system.
Comments 5: The quantitative images representing western blot doesnt seem to match up. Please check.
Response 5: Thank you for your advice. We checked the blotting again, and we would like to stay with the values. Initially, the experiment was designed to included two plant extract. That’s why the original images had more than 6 groups. The additional three blots corresponded to the other plant extract, which was not utilized in the manuscript. The replicates of the samples were supplemented in the ‘Figures, Graphics, Images’ along with the revised manuscript in the system.
Reviewer 2 Report
Comments and Suggestions for Authors
The topic of study is very interesting and of great importance in our days. The results are very interesting, just the presentation and the highlight of the proved effects and the mechanism of action must be improved.
The significance of letters from figures must be in detail presented (in the legend).
There no references in the part of methods. Are all methods originals of authors? If not, the references must be included, in text and in references list.
At discussion the authors conclude that their results of 20,6 mM MIC is similar with those already presented in other papers by other researchers that is 0,02 microM. I suggest to the authors to verify these data because the values are very different. 20.6 mM is 206000 microM. There are a few digits difference.
As overall observation is that if sophoridine has antimicrobial effect on pathogens, but how is acting on the microbial species from microbiota. Did the authors had data regarding this? To explain the anti-diarrheal effect this is very important.
Another observation and suggestion to the authors is to explain that the antimicrobial effects, the values of MIC and MIB are higher than the maximum concentration for that the sophoridine is still without cytotoxicity. Meaning that the concentration of sophoridine needed to exhibit antimicrobial effect will have also cytotoxic effect.
Comments on the Quality of English Language
There are many typing errors that must be corrected. E.g at chapter 2.1 an entire fragment of text is doubled.
The English language must be improved, there are expression mistakes that must be corrected or reformulated.
Author Response
Reviewer 2
Comments 1: The topic of study is very interesting and of great importance in our days. The results are very interesting, just the presentation and the highlight of the proved effects and the mechanism of action must be improved.
Response 1: Thank you for your advice. We modified the presentation and the highlight of proved effects and mechanism of action in the conclusions (Line 427-435) of the revised manuscript.
Comments 2:The significance of letters from figures must be in detail presented (in the legend).
Response 2: Thank you for pointing this out. The different letters denote the significance between groups. Specifically, ‘a’ represents the highest value, and the value decrease in sequence with ‘b’ ‘c’ or ‘d’. when two groups have the same letter, there is no significantly difference between them. Conversely, if two groups have different letters, a significant difference exists.
Comments 3:There no references in the part of methods. Are all methods originals of authors? If not, the references must be included, in text and in references list.
Response 3: Thank you for pointing this out. Some of the testing indices were measured by testing kits according to the protocol provided by the manufacture’s instruction, and we provided the references for relevant methods in the text and references in Line 307, 323, 390, 406, 408 and 590-599 of the revised manuscript.
Comments 4:At discussion the authors conclude that their results of 20,6 mM MIC is similar with those already presented in other papers by other researchers that is 0,02 microM. I suggest to the authors to verify these data because the values are very different. 20.6 mM is 206000 microM. There are a few digits difference.
Response 4: Thank you for bringing this to our attention. It’s my fault to write it incorrectly. The original information in Xia’s article was 2×10-2mol/L, which is equivalent to 20mmol/L. Our result was 20.6mmol/L, showing a similarity. And we corrected it in Line 190-191 of the revised manuscript.
Comments 5:As overall observation is that if sophoridine has antimicrobial effect on pathogens, but how is acting on the microbial species from microbiota. Did the authors had data regarding this? To explain the anti-diarrheal effect this is very important.
Response 5: Thank you for your advice. We indeed possess data on intestinal microbiota, which will be analyzed in a forthcoming paper in the near future. In the current study, we found that a sophorodine concentration exceeding 2.56 mg/mL exhibited no cytotoxicity. However, 2.56 mg/mL was significantly lower than the inhibitory and bactericidal concentrations of sophorodine against the several bacteria examined in this study. Consequently, the antibacterial effect of sophorodine within the intestine can be ruled out. We do not believe that its antidiarrheal effect is related to its antibacterial property; rather, it is associated with its anti - inflammatory effect. This implies that sophorodine may act on intestinal tissue rather than on intestinal microbes.
Comments 6:Another observation and suggestion to the authors is to explain that the antimicrobial effects, the values of MIC and MIB are higher than the maximum concentration for that the sophoridine is still without cytotoxicity. Meaning that the concentration of sophoridine needed to exhibit antimicrobial effect will have also cytotoxic effect.
Response 6: Thank you for your question. Let me explain it. We explored the potential of sophoridine as an antidiarrheal agent in animals. The safe concentration of sophoridine was the primary factor we considered, and this was determined through the cytotoxicity test. Indeed, the concentration of sophoridine required to exert an antimicrobial effect would also have a cytotoxic effect. Although sophoridine demonstrated antibacterial activity in vitro, we did not observe any such antibacterial effect in our in vivo study. When used in animals, its antidiarrheal effect was not attributed to its antibacterial properties but rather to its anti - inflammatory effects. This was stated in the conclusion as follows: " sophorodine demonstrated potent anti-bacterial activity against several Gram-negative and Gram-positive bacteria in vitro. Sophorodine was non-cytotoxic and effectively inhibited NO production in LPS-stimulated RAW264.7 cells at concentrations below 2.56mg/mL. Additionally, it exerted a protective effect on intestinal inflammation by reducing the fecal occult blood, modulating the secretion of inflammatory factors (decreasing pro-inflammatory factors and potentially increasing anti-inflammatory ones), and enhancing the level of certain regulatory factor. "
Comments 7:Comments on the Quality of English Language. There are many typing errors that must be corrected. E.g at chapter 2.1 an entire fragment of text is doubled. The English language must be improved, there are expression mistakes that must be corrected or reformulated.
Response 7: Thank you for pointing this out. The mistakes occurred in chapter 2.2 when we tried to put the figure and the figure title and annotation in the same page through multiple pasting and copying operations, and we deleted the repetitive parts of the manuscript and corrected title of 4.2 in the manuscript. Regarding the English language should be polished, we have indeed made improvements, and all the modification were marked in the revised manuscript.
Round 2
Reviewer 1 Report
Comments and Suggestions for Authors
None.